# Preparation Method and Application of Nanobubbles: A Review

**Yanwei Wang * and Tianxiang Wang**

School of Mechanical Engineering, Heilongjiang University of Science & Technology, Harbin 150022, China; wtx2548268044@163.com
* Correspondence: wangyanwei@usth.edu.cn

**Abstract:** Nanobubbles represent a special colloidal system, as they have high stability and large specific surface areas. The preparation of nanobubbles is currently a hot research topic, as it crucial to investigate their characteristics and expand their applications. This article explains the mechanism of generating nanobubbles based on chemical and physical methods, introduces their basic composition's structure and properties, summarizes the methods of preparing bulk nanobubbles (BNBs) and surface nanobubbles (SNBs), and clarifies the preparation principles and techniques. Seven practical applications of nanobubbles are cited in this paper, including their use as ultrasonic contrast agents in medical imaging, drug delivery systems in drug transportation, promoters of plant growth by affecting plant respiration and water absorption at the roots, tools to remove dirt from surfaces by generating energy during nanobubble bursting, producers of high-density negative ions and free radicals to react with pollutants in wastewater, tools to reduce the resistance of the fluid flow through channels by lowering the internal friction, and means of improving the mineral flotation recovery rate by enhancing the absorption capacity of bubbles to minerals. Finally, the future development of nanobubble preparation technology is discussed, including their roles in optimizing equipment and preparation methods; improving the quantity, efficiency, stability, controllability, and homogeneity of nanobubble generation; and promoting the industrial production of nanobubbles.

**Keywords:** nanobubbles; colloidal system; preparation method; generation mechanism; contrast agent; drug transport





## 1. Introduction

Bubbles are commonly found in nature and typically refer to spherical or hemispherical bodies that form on the surface of a liquid due to gas dispersion. In engineering, they are usually formed by introducing gas into a liquid layer through small apertures, but they exhibit low stability and are prone to explosion [1]. Bubbles are commonplace in daily life, such as soap bubbles formed via rubbing during laundry, bubbles generated during soup boiling, and water bubbles formed when raindrops hit the surface of a body of water. Although scientists have thoroughly studied bubbles visible to the naked eye, nanobubble research is still ongoing. As research into nanobubbles deepens, it has been discovered that compared to larger bubbles, nanobubbles exhibit ultra-high stability and represent a unique colloidal system [2]. The lifespan of bubbles commonly found in daily life is typically a few seconds to several minutes, while nanobubbles can survive for several days and even stably exist for over a month under certain conditions [3]. Nanobubbles are categorized into surface nanobubbles and bulk nanobubbles, as illustrated in Figure 1. Surface nanobubbles refer to nanoscale bubbles attached to solid surfaces at solid–liquid interfaces, which typically have cap-like shapes. The contact line radius of surface nanobubbles generally ranges from 50 to 500 nm, as well as heights between 10 and 100 nm. Bulk nanobubbles represent spherical bubbles uniformly dispersed in water or other liquids, which have diameters smaller than 1000 nm [4].

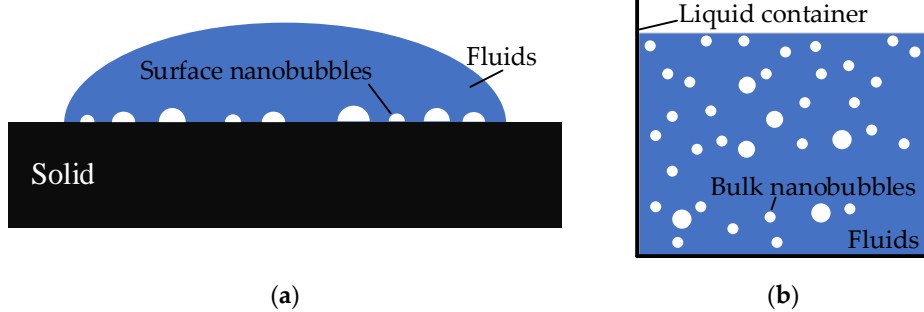

**Figure 1.** Surface and bulk nanobubbles: (**a**) surface nanobubbles; (**b**) bulk nanobubbles.

Currently, nanobubbles have widespread applications in various fields, including medicine [5], agriculture [6], and industry [7]. In the medical field, nanobubbles are often utilized in medical imaging [8] and drug delivery [9] due to their excellent contrast and material transport capabilities. In the agricultural sector, nanobubbles are often utilized to promote plant growth by enhancing water permeability [10]. In industry, nanobubbles not only enhance a solution's oxidation capacity, but also undergo chemical reactions with pollutants; therefore, they are widely applied in wastewater treatment [11] and surface cleaning [12] fields. With the development of nanobubble research and the improvement in preparation technology, nanobubbles will become more widely used. Therefore, developing and improving methods of producing efficiently and stably nanobubbles will become increasingly important. Robust methods used for the large-scale production of nanobubbles can provide a vital basis for their application in fields such as medicine, agriculture, and industry. This review introduces the current preparation methods and applications of both BNBs and SBNs, with the hope of raising awareness among researchers and encouraging them to engage in nanobubble research, thereby promoting its application in various fields. With a deeper understanding of the properties and applications of nanobubbles, we believe that this field will become extremely important, providing a more effective means of solving various real-world problems.

## 2. The Composition and Stability of Nanobubbles

### 2.1. The Composition of Membrane Nanobubbles

Membrane nanobubbles consist of three basic elements: a gas core, a shell layer, and a liquid phase. The gas core comprises gas, while the shell layer is primarily composed of surfactants, polymers, or lipids. The liquid phase consists of water and either inorganic salt solutions or organic small-molecule solvents [13], as illustrated in Figure 2a. The stability of membrane nanobubbles primarily depends on the shell layer, wherein a dense molecular layer can protect the gas core and slow down its diffusion [14].

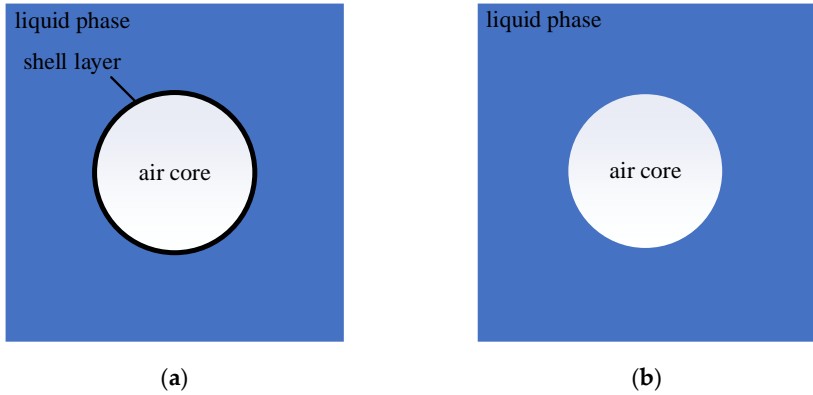

**Figure 2.** Nanobubble structure diagram: (**a**) membrane nanobubbles; (**b**) non-membrane nanobubbles.

## 2.2. The Composition of Non-Membrane Nanobubbles

Non-membrane nanobubbles are composed of two basic elements: a gas core and a liquid phase [14], as illustrated in Figure 2b. The classic Epstein–Plesset theory predicts that the smaller the size of the filmless bubble, the shorter its life expectancy. However, researchers have found that non-membrane nanobubbles exhibit high stability, even without a membrane [15].

## 2.3. The Stability of Nanobubbles

The stability of nanobubbles refers to their ability to persist under specific conditions. Various factors influence their stability, including the surface charge of the bubbles, surface tension, and properties of the solution. The primary reason for the stability of nanobubbles lies in their possession of surface charge [16]. Nanobubbles with a high zeta potential exhibit greater stability. However, the zeta potential of nanobubbles decreases over time. Although gas diffusion suggests that the bubble size should decrease, research indicates that the bubble size actually increases. This phenomenon is attributed to the combined effects of zeta potential reduction and Brownian motion, causing bubble coalescence to occur and larger bubbles to be formed [17].

## 3. Preparation Methods of BNBs

BNBs can be formed in a liquid by manipulating gas pressure, ultrasonic intensity, or stirring intensity. Common methods used to prepare BNBs include mechanical stirring, gas dissolution release, pressure variation, and cavitation. In addition, microfluidic and nanoporous membrane methods can be used to prepare BNBs. This chapter provides an introduction to the methods of preparing BNBs and concludes by providing a summary of the advantages and disadvantages of each method, as shown in Table 1.

**Table 1.** The advantages and disadvantages of the methods of preparing bulk nanobubbles.

| Methods | Advantages | Disadvantages |
|---|---|---|
| Mechanical Stirring Method | The principle is simple and easy to implement | Only a small number of nanobubbles can be prepared |
| Nanoscale pore membrane method | Enables control over bubble size and distribution | Requires specialized membranes with accurate pore sizes. Potential blockage or fouling of pores may reduce efficiency over time |
| Microfluidic method | Enables precise control of bubble size and distribution. Offers a high degree of automation and integration with other processes | Requires complex microfluidic devices and fabrication techniques |
| Acoustic cavitation method | Efficient and rapid generation of nanobubbles | Requires specialized equipment and ultrasound sources. Control over bubble size and distribution may be limited |
| Hydrodynamic cavitation method | High energy efficiency, low cost, and scalability | Efficiency can be influenced by factors such as the flow rate and pressure. |
| Dissolved gas release method | Easy and straightforward to implement. Low cost | Limited control over bubble size and distribution. May result in larger bubble sizes compared to other methods |
| Periodic pressure variation method | A more uniform bubble can be prepared, and the size of the bubble can be controlled by controlling the pressure and period. | Only a small number of nanobubbles can be prepared |
| Hydraulic air compression method | Nanobubbles can be produced on a large scale at low cost and with high efficiency. | Limited control over bubble size and distribution |

### 3.1. Mechanical Stirring Method

The goal of the preparation of BNBs using mechanical agitation involves subjecting a liquid phase containing surfactants to iterative rotational stirring through a mechanized mechanism. The resultant high shear, intense turbulence, collision effects, and hydrodynamic cavitation induced during agitation facilitate interactions between the gas and liquid phases, leading to bubble generation. These bubbles, which are subjected to multiple cycles of agitation, undergo continuous shearing, resulting in the formation of progressively smaller bubbles, ultimately giving rise to BNBs [18]. Etchepare et al. [19] conducted experiments regarding the preparation of BNBs using the mechanical stirring method, using the experimental setup shown in Figure 3. BNBs were formed using a pump and circular column under various pressures and air–liquid interfacial tensions. The results showed that this method can rapidly generate BNBs, which can remain stable for over 60 days. Senthilkumar et al. [16] used mechanical stirring to generate nanobubbles in heat transfer oil. The results showed that the nanobubbles generated had diameters of less than 200 nm, and their presence was able to improve the thermal conductivity and viscosity of the heat transfer oil. Jadhav et al. [20] conducted a study using various hollow-shaped rotating mechanisms to explore the impact of different hollow shapes, rotational speeds, operating times, and temperatures on the generation of BNBs in pure water. The results revealed that while density significantly varied across different hollow shapes, the size distribution, average bubble diameter, and zeta potential remained relatively consistent. Moreover, increasing the rotational speed, prolonging the operating time, and raising the temperature allowed the generation of higher concentrations of bubbles, as these actions facilitated the release of more air from the water.

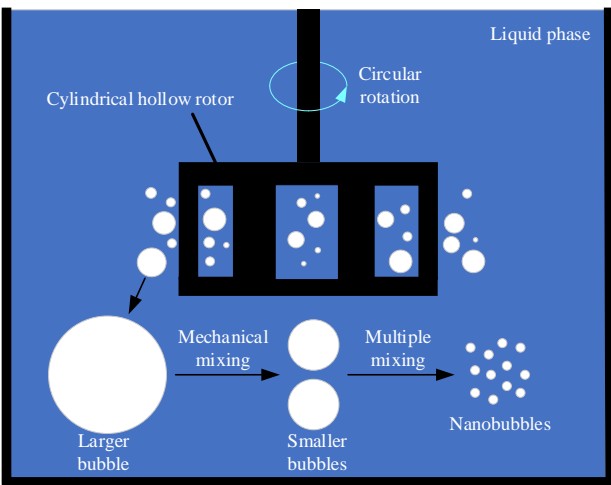

**Figure 3.** Schematic of the goal of the mechanical stirring method for nanobubble preparation.

### 3.2. Nanoscale Pore Membrane Method

The goal of preparing BNBs via the nanoporous membrane method is to press gas into the nanoscale pores of the membrane under a certain pressure. At the beginning of gas entering the liquid phase, the diameters of the nanobubbles are the same as that of the pore. However, as the BNBs expand, the diameters of the nanobubbles gradually increase, and the drag force caused by the water flow also increases. Under the action of the drag force, the nanobubbles will detach from the pore, obtaining BNBs larger than the pore diameter. Additionally, the smaller the liquid flow rate, the smaller the size of the resulting BNBs [21]. The process of generating BNBs is shown in Figure 4. The SPG (Shirasu Porous Glass) membrane is a new type of inorganic membrane developed by SPG Corporation, Japan, in 1981. SPG membrane has a uniform and uniform micropore size, and the size of pore is easily changed. Kukizaki et al. [22] used SPG membranes with nanoporous membranes to prepare nanobubbles. The experimental setup is shown in Figure 5. Air was compressed and introduced into a sodium dodecyl sulfate solution with a concentration

ranging from 0.05 to 0.5 wt.%. The solution was then passed through a SPG membrane, which had a transmembrane/bubble point pressure ratio of 1.1–2.0. Under these conditions, monodisperse nanobubbles with average diameters of 360–720 nm were stably prepared. The average diameter of the resulting BNBs was found to be 8.6 times larger than that of the pore, and its size was not significantly affected by the air velocity and liquid surface tension. Therefore, the size of BNBs could be controlled by adjusting the pore size of the membrane. Ahmed et al. [23] used tube ceramic membranes to prepare BNBs and found that the size of BNBs could be directly influenced by injecting air at different pressures into the water through the tube. The size of BNBs under the same injection pressure was related to the pore size of the tube ceramic membrane, which was similar to the experimental result of Kukizaki et al. [22]. Zhang et al. [24] developed a membrane-based physical sieving method to prepare BNBs of controllable sizes. The goal of this method is to adjust the size range of generated BNBs by controlling the gas filtration rate and the quality of the membrane. BNBs sieving experiments were conducted using three types of membranes, and the results showed that the membrane could not only crush larger bubbles into smaller bubbles, but also fuse small bubbles into larger bubbles when filtering BNBs.

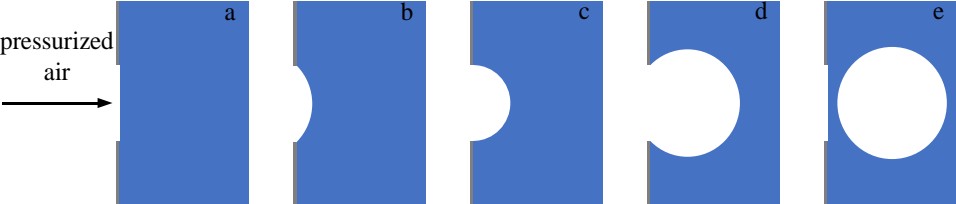

**Figure 4.** Process of BNB generation via the membrane method: (**a**) initial state; (**b**) preliminary growth stage of nanobubbles; (**c**) nanobubbles grow to a diameter equal to that of the pore; (**d**) continual growth stage of nanobubbles; (**e**) detachment of nanobubbles.

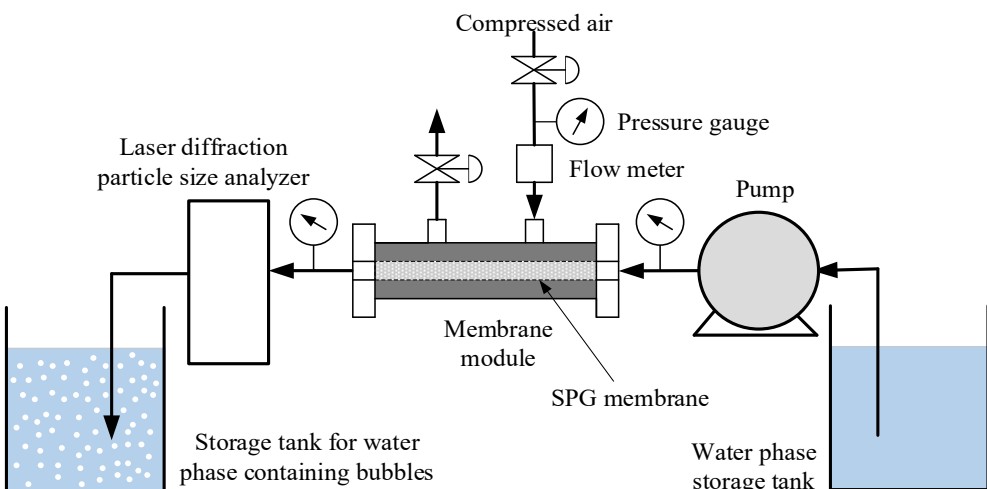

**Figure 5.** Schematic diagram of the experimental setup used for BNB preparation via the SPG membrane method [22].

### 3.3. Microfluidic Method

The goal of the preparation of BNBs via microfluidics involves the utilization of microfluidic chips to regulate the flow of mixed gas and liquid [25]. The gaseous mixture is introduced through a gas inlet and, when passing through the liquid phase, it experiences viscous forces exerted by the liquid, resulting in the formation of microbubbles. Some of the gas within these microbubbles dissolves into the aqueous phase and eventually shrinks, giving rise to BNBs. Xu et al. [26] were the first group of researchers to utilize a microfluidics-based approach for the preparation of BNBs, using the experimental setup illustrated in Figure 6. This method employs a mixed gas composed of water-soluble nitrogen and

water-insoluble perfluorocarbon (PFC) as the gaseous phase for the microfluidic bubble generator. Initially, monodisperse microbubbles are generated, which gradually shrink during the dissolution process of the water-soluble nitrogen, ultimately resulting in BNBs of a certain size. The degree of bubble contraction can be controlled by adjusting the ratio of water-soluble nitrogen and water-insoluble PFC. The most significant advantage of using this method is its precise control over the size and uniformity of the resulting BNBs.

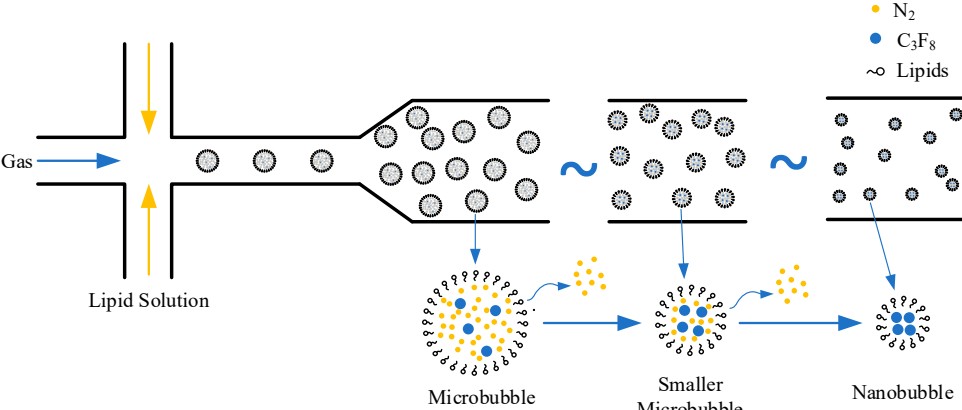

**Figure 6.** Schematic diagram of the experimental setup used for the preparation of BNBs via microfluidics.

### 3.4. Acoustic Cavitation Method

The goal of BNB preparation via the acoustic cavitation method is to induce local negative pressure in the liquid by means of a high-speed propeller rotation or negative pressure half-cycle generated via high-intensity sound waves, ultimately leading to the formation of micro- and nano-scale bubbles near tiny gas nuclei [27]. Nirmalkar et al. [28] conducted BNB preparation experiments via the acoustic cavitation method, using the experimental setup depicted in Figure 7. The study found that BNBs existed in pure water but not in organic solvents, and they disappeared at a certain ratio of organic solvent to water. This outcome is attributed to the electrostatic charge on the surface of the BNBs, which stabilizes them via the adsorption of hydroxyl ions produced via water's auto-ionization. Pure organic solvents, however, do not undergo auto-ionization.

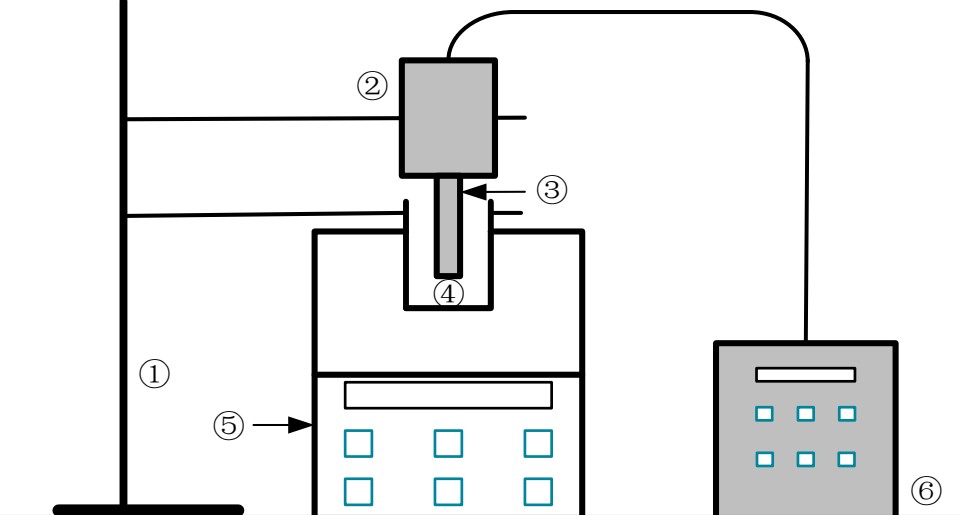

**Figure 7.** Schematic diagram of the experimental setup used for BNB preparation via the acoustic cavitation method: ① retort stand and clamps; ② ultrasonic transducer; ③ titanium probe; ④ glass beaker; ⑤ recirculating cooler; ⑥ ultrasound processor [28].

### 3.5. Hydrodynamic Cavitation Method

The hydrodynamic cavitation technique boasts several advantages, including high energy efficiency, low cost, and scalability. Its goal is to induce cavitation in a medium by altering the flow velocity of the medium, thereby causing pressure fluctuations. This outcome is analogous to that produced via acoustic cavitation techniques [29]. Therefore, hydrodynamic cavitation can be utilized instead of acoustic cavitation to generate nanobubbles. Alam et al. [30] conducted an experiment regarding the preparation of nanobubbles via hydrodynamic cavitation. The experimental apparatus employed was a two-chambered swirling jet nozzle, which generated nanobubbles in a saturated or supersaturated solution via a circulation system, as illustrated in Figure 8. The results indicated that the device successfully produced nanobubbles with diameters of less than 200 nm, and these nanobubbles carried a negative charge when present in water. Wu et al. [31], in their study, refined the cavitation reactor, employing numerical simulation to investigate the influence of various geometric parameters on the flow field structure. They successfully identified the optimal design, subsequently fabricating a laboratory-scale vortex-type micro-nanobubble generator. Flow experiments were conducted, resulting in the generation of bubbles with diameters as small as 301 nm. This endeavor provided valuable insights into the exploration of the methodologies of micro-nanobubble generation and the quest for their optimal structural configuration.

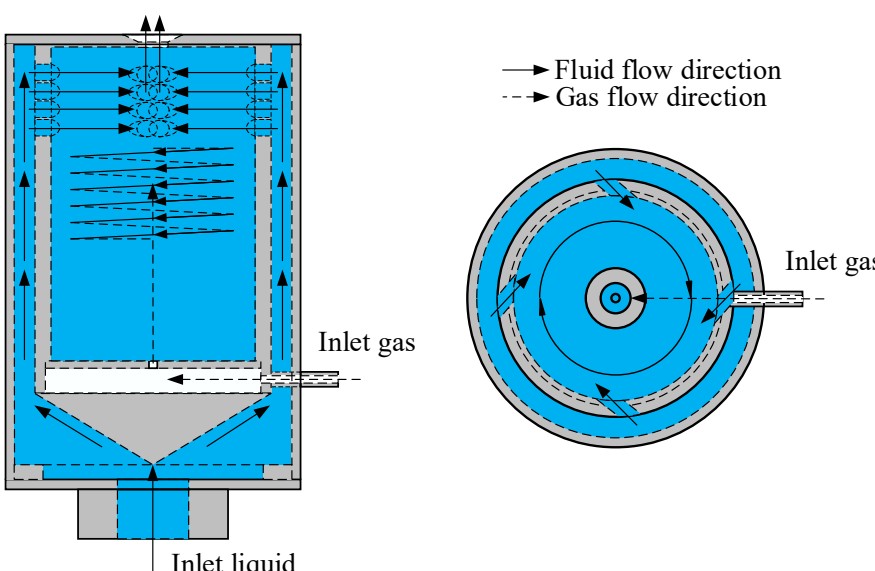

**Figure 8.** Schematic diagram of the experimental setup used for the preparation of BNBs via hydrodynamic cavitation.

### 3.6. Dissolved Gas Release Method

The goal of preparing BNBs via the gas dissolution release method is to increase the pressure available to dissolve the gas, before reducing the pressure required to make the dissolved gas molecules precipitate and form bubbles [32]. The average size of the bubbles is related to the solubility of the gas present in the solution, and the diameter of the bubbles is inversely proportional to the solubility of the gas [17]. The solubility of $CO_2$ in pure water without hydrates is inversely proportional to the temperature and directly proportional to the temperature in the presence of the hydrates [33]. Wang et al. [34] used $CO_2$ as the gas source and prepared BNBs via the gas dissolution release method. The experimental setup is shown in Figure 9. They found that the optimum gas–liquid ratio for preparing BNBs was 2.87%, the optimum operation time of the generator was 28.47 min, and the optimum inlet water temperature was 25.52 °C.

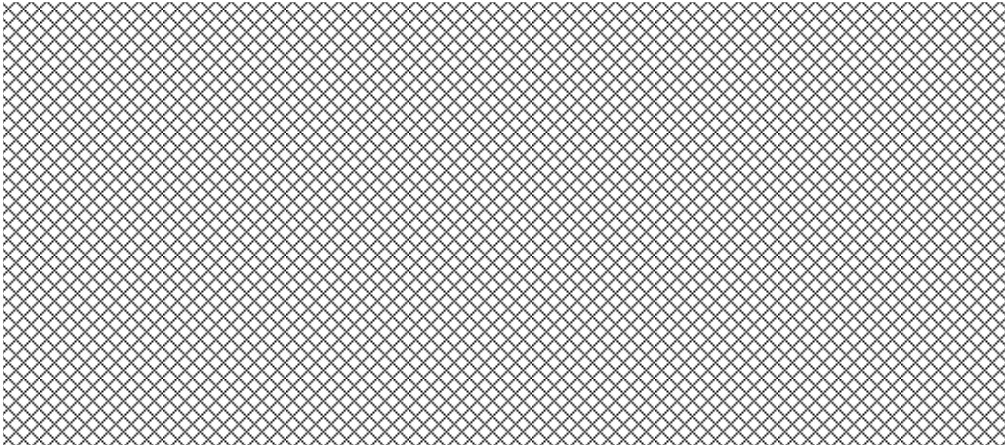

**Figure 9.** Schematic diagram of the goal of BNBs preparation via the gas release method: ① $CO_2$ gas supply bottle; ② gas flow meter; ③ diaphragm pump; ④ dissolved gas tank; ⑤ hydraulic pressure gauge; ⑥ throttling nozzle for the released gas [34].

### 3.7. Periodic Pressure Variation Method

The goal of BNB preparation via the periodic pressure variation method is controlling gas dissolution and precipitation by carrying out periodic pressure adjustments to a gas-saturated solution [35]. Wang et al. [36] utilized the periodic pressure variation method to perform BNB preparation, using the experimental setup illustrated in Figure 10. The study successfully produced stable $N_2$ nanobubbles and determined that given a constant frequency of pressure changes, longer times of exposure to pressure led to the formation of smaller nanobubbles.

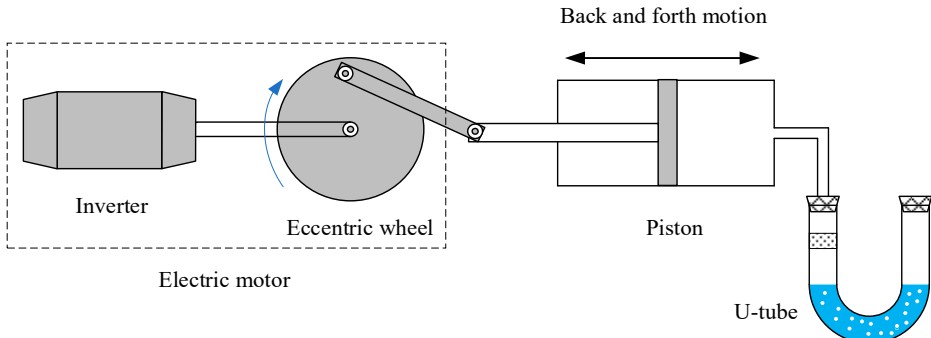

**Figure 10.** Schematic diagram of the device used for the preparation of BNBs via the periodic pressure change method [36].

### 3.8. Hydraulic Air Compression Method

Yang et al. [37] conducted an experiment regarding the preparation of nanobubbles via the hydraulic air compression method, using the experimental apparatus illustrated in Figure 11. This experiment proved, for the first time, that the hydraulic air compression method could be utilized for the generation of nanobubbles. The authors employed nanoparticle tracking analysis to assess the size distributions and concentrations of nanobubbles, and their results indicated that the concentrations of nanobubbles increased as the height of the outlet pipe increased. The hydraulic air compression method not only allows the large-scale production of nanobubbles, but is also cost-effective and highly efficient. In the future, the application of hydraulic air compression method technology may prove to be instrumental to the use of nanobubbles in industrial and agricultural settings.

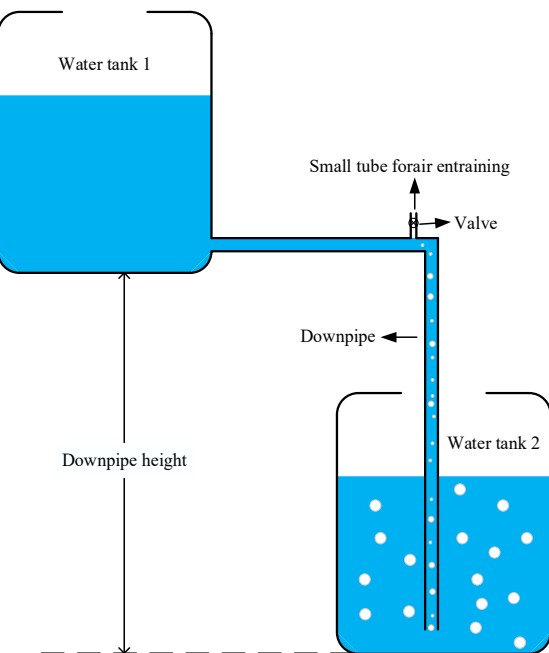

**Figure 11.** Schematic diagram of the experimental setup for the preparation of nanobubbles via the HAC method [37].

## 4. Preparation Methods of SBNs

SNBs are small bubbles that form at the interface between a liquid and a solid. Their preparation not only requires consideration of the liquid surface tension and gas solubility, but also takes into account the hydrophobicity or hydrophilicity of the solid surface, as all three factors can affect the generation and stability of SNBs. This chapter presents common methods used to prepare SNBs, including water electrolysis, the cold water method, solvent exchange, vacuum methods, and microwave radiation, and its concludes with a summary of the advantages and disadvantages of using each method, as shown in Table 2.

**Table 2.** Advantages and disadvantages of preparation methods for solid-liquid surface nanobubbles.

| Methods | Types | Advantages | Disadvantages |
|---------|-------|-----------|---------------|
| Aqueous solution electrolysis method | Chemistry | Allows precise control over the generation of nanobubbles | Requires specialized electrolysis equipment |
| Cold water method | Physics | Simple and easily accessible method. Can produce nanobubbles without the need for complex equipment | Limited control over bubble size and stability |
| Solvent exchange method | Physics | Easy and straightforward to implement. Low cost | Control over bubble size and distribution may be limited |
| Depressurization method | Physics | Allows the controlled and rapid generation of nanobubbles. | The resulting nanobubbles are unstable |
| Microwave irradiation method | Physics | No impurities are introduced, and the output of nanobubbles can be controlled | Requires specialized microwave equipment that has precise power control |

### 4.1. Aqueous Solution Electrolysis Method

Electrolysis of water produces hydrogen and oxygen molecules. When the concentration of molecules reaches the critical concentration required to carry out nucleation, bubbles will form. The average size of the bubbles generated decreases as the voltage during electrolysis increases [38]. Conductive materials, such as highly ordered pyrolytic graphite (HOPG), can be used as electrodes to obtain SNBs on their surfaces. When the

HOPG surface acts as the anode, oxygen nanobubbles are obtained. The sizes and quantities of bubbles can be controlled by adjusting the voltage and duration of electrolysis. Yang et al. used water electrolysis to prepare SNBs. The experimental setup used is shown in Figure 12; the HOPG surface was used as the electrode, and the AFM was used to detect the resulting bubbles. It was found that as the voltage increased, the volume and coverage of HPOG SNBs also increased. In addition, the yield of the oxygen nanobubbles was much lower than that of the hydrogen nanobubbles [39].

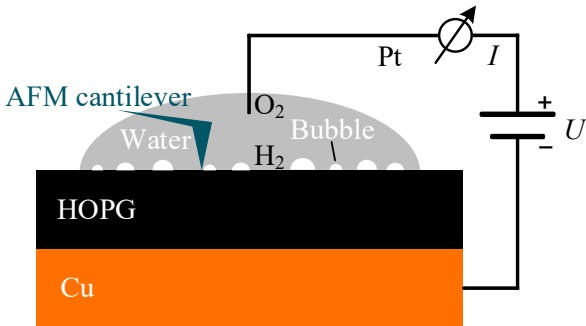

**Figure 12.** Schematic diagram of nanobubble preparation via the electrolytic process [39].

### 4.2. Cold Water Method

As per Henry's law, under constant pressure and comparable conditions, the solubility of a gas within a liquid diminishes due to an elevation in temperature [40]. The cold water method is a relatively simple approach used in the preparation of SNBs, which involves using heated graphite surfaces and cold water as the essential materials. An et al. [41] utilized the cold water method to prepare SNBs in their experiment, as illustrated in Figure 13. Specifically, they maintained pure water at a temperature of 4 °C or below for at least 12 h and baked HOPG at different temperatures (40 to 80 °C) for two hours in an oven. During the experiment, they rapidly distributed a certain amount of pure water (at 4 °C) onto the HOPG surface after removing it from the oven. Finally, they mounted the sample onto the atomic force microscopy (AFM) stage and employed AFM to carry out detection. Using multiple experiments, they detected the presence of SNBs each time and observed that as the HOPG temperature increased, the accumulation of SNBs became denser, and their lifespans lasted over five days.

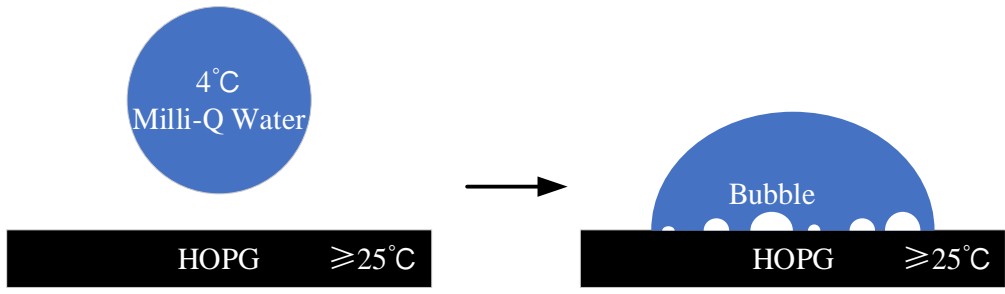

**Figure 13.** Schematic illustration of SNBS preparation via the cold water method [41].

### 4.3. Solvent Exchange Method

The solvent exchange method used to prepare nanobubbles mainly includes the ethanol–water substitution [42] and the sodium chloride solution–water substitution methods [43]. The basic principles of both methods are similar, as a liquid with high gas solubility replaces a liquid with low gas solubility. Taking the ethanol–water substitution method as an example, ethanol is first injected into the container, followed by the slow addition of pure water to the container containing ethanol. Finally, a mixed solution of ethanol and water is obtained. Qiu et al. [44] conducted a nanobubble preparation experiment using the ethanol–water exchange method, as illustrated in Figure 14. The study found that this

approach yields abundant nanobubbles, and their sizes and concentrations were measured using a nanoparticle tracking analyzer. It was observed that the higher the amount of gas dissolved in the solution, the greater the number of nanobubbles that formed.

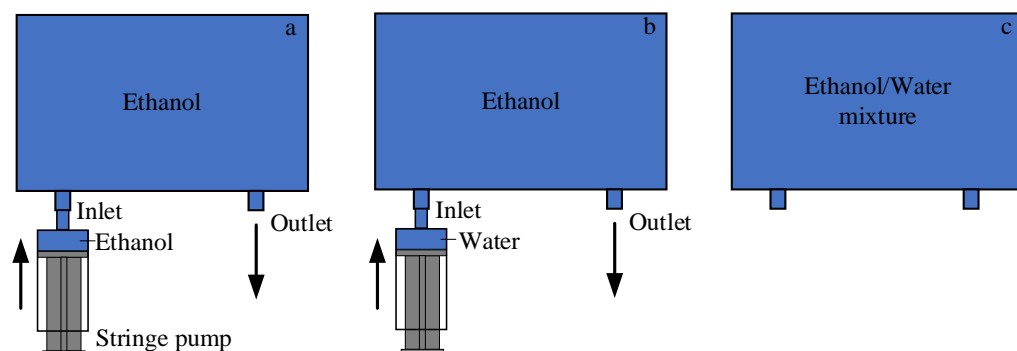

**Figure 14.** Schematic illustration of SNBs preparation via the ethanol–water solution exchange method. (**a**) Ethanol is injected into a sealed liquid container using a syringe; (**b**) Then replace the ethanol with deionized water; (**c**) Bubbles are created when ethanol replaces the water solution [44].

### 4.4. Depressurization Method

The goal of preparing nanobubbles via the depressurization method is to alter the gas solubility by controlling the pressure exerted on the liquid within the container. When the pressure is reduced, the gas solubility decreases, causing precipitation and the formation of nanobubbles [36]. Fang et al. [45] dripped unsaturated pure water onto HOPG and performed a five-minute depressurization test, which resulted in the detection of the nanobubbles through AFM. This result indicates that a short-term reduction in pressure locally saturates the gas concentration on the surface of the HOPG and generates nanobubbles. However, when the depressurization time was extended to 20 min, no nanobubbles were detected, suggesting that some gas molecules had escaped from the liquid.

### 4.5. Microwave Irradiation Method

The electromagnetic wave irradiation method provides a convenient and impurity-free approach to preparing SNBs. Its principle is based on the photons of the electromagnetic wave irradiating the hydrophobic surface of the water, where the energy-carrying photons increase the probability of gas escaping from the interface. Under microwave radiation, the solubility of the gas in water decreases, which favors the nucleation of nanobubbles. The schematic diagram of bubble generation via microwaves is shown in Figure 15. Wang et al. hypothesized the mechanism of nanobubble generation via microwave irradiation, as shown in Figure 16. They also conducted a nanobubble preparation experiment using microwave irradiation, in which they used highly pure oxygen (99.995%) as the gas source to inflate deoxygenated pure water and HOPG as the substrate. The newly cut HOPG was then fixed in the oxygenated water, and microwave radiation was applied to the solution to generate SNBs. The study found that adjusting the gas concentration, irradiation time, and working power could control the yield of the nanobubbles [46]. Yuan et al. [47] utilized accelerated electron irradiation to synthesize bulk nanobubbles (BNBs) in pure water. The outcomes demonstrate a direct correlation between nanobubble formation and the escalating irradiation dose rates. Moreover, with an elevated irradiation dose, the initial augmentation of the nanobubble concentration is followed by subsequent attenuation.

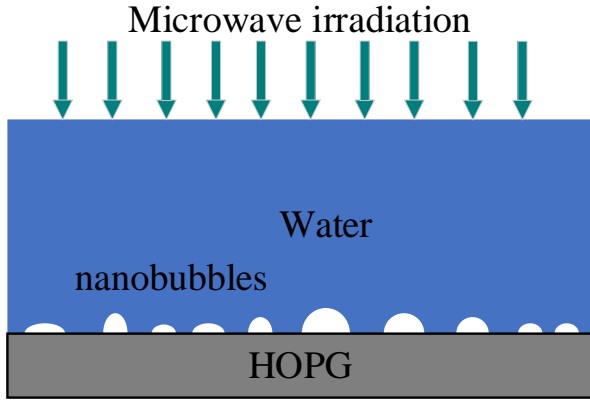

**Figure 15.** The schematic diagram of nanobubble generation via microwave [46].

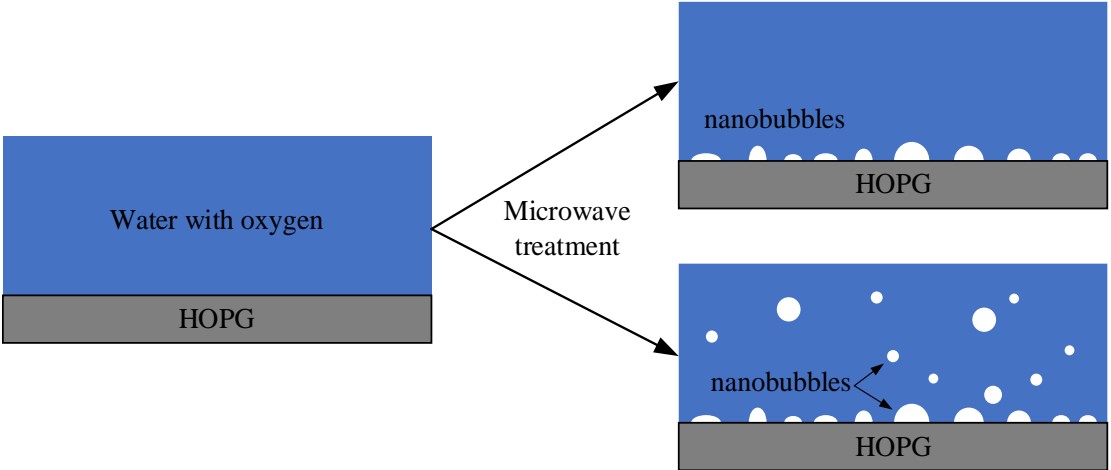

**Figure 16.** The possible mechanism of nanobubble formation [46].

There are various methods for preparing SNBs, with each method having its characteristics. Water electrolysis, which generates gas through the electrolysis reaction generated in the liquid by applying an electric field, is a simple method suitable for large-scale production, but it has some limitations in terms of the choice of electrode materials and requires energy consumption. Cold water, solvent exchange, and vacuum methods change the gas solubility by changing the solution's temperature, composition, and pressure, respectively, causing the gas to precipitate on the substrate surface to obtain SNBs. Among these methods, the cold water method is only suitable for the preparation of a small number of SNBs, while the solvent exchange and vacuum methods can prepare a larger quantity of nanobubbles. Microwave radiation induces the generation of SNBs by acting on the interface with the energy-carrying photons. The advantage of this method is that it does not introduce any impurities during the entire process.

## 5. Applications of Nanobubbles

Nanobubbles, which are also known as ultrafine bubbles, have garnered extensive application across various domains due to their distinctive properties. This chapter expounds the utilization of nanobubbles within fields encompassing medicine, agriculture, and industry, outlining specific instances of use, as depicted in Figure 17. In the realm of healthcare, nanobubbles find utility as contrast agents for ultrasound imaging and carriers for the dispensing of medical compounds. Given the medical sector's stringent requirements regarding nanobubble size distribution, their preparation necessitates the employment of microfluidic techniques and nanoporous membrane methodologies. In the agricultural sphere, nanobubbles have the ability to stimulate plant growth. As preci-

sion regarding nanobubble dimensions and distribution is less significant in agriculture, relatively straightforward mechanical agitation methods can be employed. Within the industrial milieu, nanobubbles not only enhance wastewater treatment and surface cleansing efficiency, but also find application in the mitigation of channel resistance and foam flotation processes. Diverse circumstances within the industrial domain warrant distinct nanobubble preparation techniques, as outlined in Table 3.

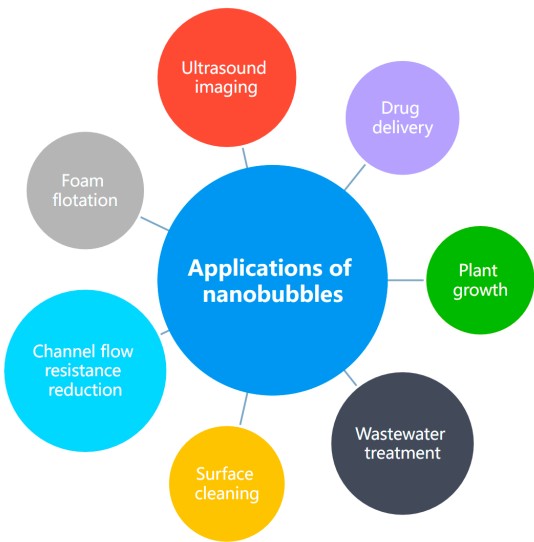

**Figure 17.** Application areas of nanobubbles.

**Table 3.** Methods used in the field of nanobubble applications.

| Application Areas | Applicable Nanobubble Preparation Methods |
|---|---|
| Ultrasound imaging | Microfluidic method<br>Nanoscale pore membrane method |
| Drug delivery | Microfluidic method<br>Nanoscale pore membrane method |
| Plant growth | Mechanical stirring method |
| Wastewater treatment | Acoustic cavitation method |
| Surface cleaning | Mechanical stirring method |
| Channel flow resistance reduction | Aqueous solution electrolysis method<br>Microwave irradiation method |
| Foam flotation | Acoustic cavitation method |

*5.1. Ultrasound Imaging*

Ultrasound imaging is a medical tool that can diagnose various diseases, and it has several advantages, such as high biological safety, free availability, dynamic observation, and real-time detection. It plays an extremely important role in medical diagnosis, being one of the most important techniques used in medical diagnoses [48]. In the ultrasound field, nanobubbles undergo alternating compression and expansion, generating unique echo signals. These echo signals are stronger than the signals from surrounding tissues; hence, nanobubbles are frequently used as contrast agents in ultrasound imaging. Compared to other ultrasound contrast agents, nanobubbles have smaller sizes and higher stability, making them better able to penetrate the blood vessels' walls and enter specific regions to carry out imaging [49]. Liu et al. [50] prepared nanobubbles containing two consecutive lipophilic dyes by combining fluorescence resonance energy transfer (FRET) and bioluminescence resonance energy transfer (BRET). They found that the microvasculature of perfused tissue could be depicted using high spatial resolution through ultrasonic

contrast-enhanced imaging using BRET–FRET nanobubbles. This BRET–FRET nanobubble contrast agent was also applied to the imaging of breast cancer animal models.

### 5.2. Drug Delivery

The high mass transfer efficiency of nanobubbles confers upon them the ability to act as therapeutic carriers to carry out drug delivery in medicine. Nanobubbles exhibit magnetic responsiveness under the action of a magnetic field and explosive behavior under ultrasound energy irradiation, which can promote self-destruction and changes in cell membrane permeability. These prerequisites are needed to achieve targeted drug delivery using nanobubbles, as illustrated in Figure 18. Sanlıer et al. [9] developed a novel dual-drug system using nanobubbles to carry out targeted therapy against non-small cell lung cancer, which is a new type of magnetic-targeting and ultrasound-responsive system. The release of drugs from the nanobubbles is achieved under the action of a magnetic field and ultrasound energy in order to achieve therapeutic effects. Cavalli et al. [51] prepared chitosan nanobubbles using perfluoropentane core and chitosan shell, enhancing the stability of the nanobubbles by adding polyvinylpyrrolidone. These nanobubbles possess excellent oxygen-carrying capacity, are non-toxic to cells, and can continuously release oxygen when injected into a low-oxygen solution following oxygenation preparation.

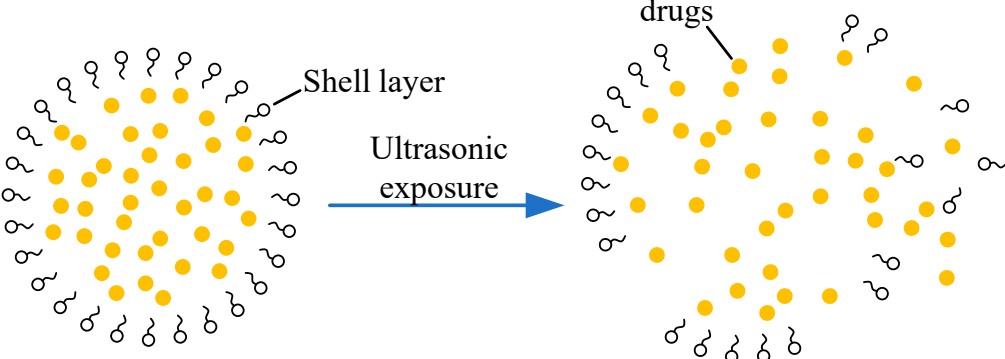

**Figure 18.** Nanobubble drug release process.

### 5.3. Plant Growth

Nanobubbles have significant impacts on plant growth, as water containing nanobubbles can provide more dissolved oxygen to plants. Higher levels of oxygen promote plant respiration, while nanobubbles can increase the surface tension between water molecules, making it easier for water to penetrate into the soil and, thus, improving the absorption of water by plant roots [52]. On the other hand, nanobubbles possess high surface energy and active sites, which enable them to disrupt the external structures of pathogenic bacteria upon contact, achieving a bactericidal effect and reducing the risk of plants contracting diseases [10]. Liu et al. [53] conducted seed germination experiments using nanobubbles, and the results showed that water containing nanobubbles had a promoting effect on seed germination.

### 5.4. Wastewater Treatment

Nanobubbles contained in wastewater can enhance the efficiency of oxidants and cleaning agents. Moreover, the relatively high concentrations of negative ions and free radicals generated by nanobubbles can undergo chemical reactions with organic compounds or heavy metals in wastewater, achieving the desired effect of treating wastewater [54]. Ahmadi et al. investigated the effects of nanobubbles on oxygen transfer and sludge yield in a sequencing batch reactor used to carry out activated sludge. The results showed that nanobubble aeration increased the oxygen content in the reactor solution, improved the oxygen transfer efficiency, and significantly reduced sludge production [11].

### 5.5. Surface Cleaning

Nanobubbles are capable of having stable existence in liquids and possess small size and high surface area characteristics. These characteristics enable them to effectively penetrate the crevices on surfaces, and when nanobubbles burst, they release energy that forms tiny liquid jets on the surface. These liquid jets can effectively remove surface dirt and impurities [55]. Wu et al. [12] proposed a method to remove dirt using nanobubbles formed via electrochemical methods, and the results showed that nanobubbles can significantly reduce surface dirt.

### 5.6. Channel Flow Resistance Reduction

Nanobubbles can form a stable dispersion system in fluids, which can reduce the viscosity and internal friction of the fluid, thereby reducing resistance. Moreover, nanobubbles can form a protective film on the surfaces of pipes or equipment, reducing the solid–liquid contact area during flow and, thus, reducing flow resistance [56]. Gao et al. [57] generated nanobubbles on the surface of polydimethylsiloxane microchannels using a vacuum method, and they quantitatively measured the slip length of the bubble surfaces in the microchannels using particle image velocimetry. Their research found that the flow boundary conditions on the surface covered by bubbles transitioned from no slip to slip, resulting in a significant drag reduction effect.

### 5.7. Foam Flotation

Froth flotation is an important mineral separation technology that uses the adhesion and lift of bubbles in solution to separate target minerals from other impurities [58]. Nanobubbles are widely used in froth flotation due to their large surface areas and stability, which can improve the adsorption capacity of bubbles found on mineral particles and increase their adhesion to the mineral surface [59]. Nanobubbles are also stable, meaning that they are less likely to burst and disappear during the flotation process, which ensures the effectiveness of froth flotation [2]. Chen [60] studied the effect of nanobubble pre-treatment on the flotation of fine dolomite and found that the recovery rate of fine dolomite increased by 7%, while the reagent dosage was reduced by 25% under the same recovery rate. Zhang et al. [61] studied the effects of nanobubbles on the flotation of ultrafine graphite and found that compared to traditional flotation, nanobubble flotation increased the average size of hydrophobic aggregates of ultrafine graphite, effectively recovered finer graphite particles, and increased the recovery rate of ultrafine graphite.

## 6. Summary and Outlook

In recent years, the practicality of nanobubbles has been demonstrated in many fields. Their high stability, mass transfer efficiency, large surface area, strong controllability, drug loading capacity, and good biocompatibility have shattered traditional perceptions of bubbles. In the field of agriculture, nanobubbles can promote the absorption of water and nutrients, increase crop yields, and reduce the usage of chemicals. In the medical field, nanobubbles have the potential to be used in advanced drug delivery systems, ultrasound imaging enhancement, and targeted therapy. Furthermore, researchers are exploring the uses of nanobubbles in environmental restoration areas, such as wastewater treatment, groundwater purification, and soil remediation. In summary, nanobubbles hold immense potential for use in a wide range of applications, revolutionizing various industries and contributing to sustainable solutions.

This article introduces the specific applications of nanobubbles in various fields, providing an overview of the principles and methods used to prepare both BNBs and SNBs. These methods can be selected and optimized according to specific application requirements. Despite the significant domestic and international progress made in the creation of technology to prepare nanobubbles, most methods yield a relatively small quantity of nanobubbles, and their size lacks uniformity, making large-scale production challenging. Future research can delve deeper into methods used to improve nanobubble

stability, controllability, uniformity, and production efficiency. For instance, improvements can be made to the structures of preparation devices, the optimization of the bubble generation process, and the exploration of new material combinations that are more suitable for use in bubble generation, thereby enhancing their application in agriculture, industry, medicine, and other fields. Additionally, the scope of nanobubble applications can be expanded to include areas such as energy storage and catalytic reactions in the energy sector, aiming to improve energy conversion efficiency and promote sustainable development.

**Author Contributions:** Data curation, Y.W.; writing—original draft preparation, T.W.; writing—review and editing, Y.W.; visualization, T.W.; supervision, Y.W. All authors have read and agreed to the published version of the manuscript.

**Funding:** This research received no external funding.

**Institutional Review Board Statement:** Not applicable.

**Informed Consent Statement:** Not applicable.

**Data Availability Statement:** Not applicable.

**Conflicts of Interest:** The authors declare no conflict of interest.

## Abbreviations

| | |
|---|---|
| BNBs | Bulk nanobubbles |
| SNBs | Surface nanobubbles |
| SPG | Shirasu Porous Glass |
| PFC | Perfluorocarbon |
| HOPG | Highly ordered pyrolytic graphite |
| AFM | Atomic force microscopy |
| FRET | Fluorescence resonance energy transfer |
| BRET | Bioluminescence resonance energy transfer |

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
