# Peer review of "Preparation Method and Application of Nanobubbles: A Review"

_coatings, doi:10.3390/coatings13091510_

Round 1

Reviewer 1 Report

This current review manuscript explained the generation mechanism

of nanobubbles based on chemical and physical methods, introduces their basic composition structure and properties, summarizes the preparation methods of bulk nanobubbles (BNBs) and surface nanobubbles (SNBs), and clarifies the preparation principles and techniques. I think it can be published after revision.

1. Please compare the advantages and disadvantages of each preparation method.

2. In the section "Applications of nanobubbles". Please add two informative figures for better presentation of this section.

3. Please add a general table including method preparation and applications of nanobubbles.

4. Please add a new section regarding limitation of nanobubbles

Please elaborate the section "Summary and Outlook ''

5. Please use of new references (2022 and 2023)

Reviewer 2 Report

Point 1:- in line 86-99, you had just mentioned the superficial mechanism and in examples also there is no mention of any specific material used. Try to explain the exact mechanism , how the stirring creates the bubbles and what is material leads to bubble formation and how stable they are and what are the variables they can effect the bubble formation. Examples should be given with full specification. 

Point 2:- In figure 1, is mechanical stirring act as propeller and do any specific shape of propeller important for bubble formation. How stable they are 

Point 3:- in line 122, you had give example with no specification of material used, what pressure is required is not mentioned

point 4:- In figure 4, what is the basis where small bubbles don't go for coalescence , is any other material is added to prevent such instability 

Point 5:- In line 135, explain all forces are involved in Microfluidic method. as they have impact in bubble breakdown such cavitation force etc 

Point 6:- line 209, write properly the style of heading

Point 7:- Mention table column where different methods of bubbles creation based their application as you had mentioned different method of bubbles preparation but which method to be used in field of application.

Point 8:- wherever you method different method try to explain with examples.

Point 9:- list of abbreviation 

Point 10:- do have draw the figures draw by self , do you needed to copyright permission 

Reviewer 3 Report

The whole manuscript should be checked carefully for spelling and stylistic errors. Some long sentences, misspellings, etc., are still noticeable throughout the text. Please carefully proofread the manuscript to ensure that it is free from any errors.

Reviewer 4 Report

Attachment provided with review comment.

Moderate editing of English language required

Reviewer 5 Report

The paper structure is adequate and well-written. The research needs and importance have been well justified in the introduction. And the conclusions are well supported by the results. I recommend publication after minor revision. Attention should be paid to the following minor aspects before publication.

- I think it would be nice to include a section comparing nanobubbles with other forms of nanomaterials such as nanotubes. I recommend referring to the papers [doi.org/10.1016/j.supflu.2018.03.007 and doi.org/10.1016/j.talanta.2018.10.047]. Afterwards, it would be interesting to make a comparison with other forms such as nanospheres [doi.org/10.1021/nn406590q]. The comparison with this type of shape is closer to the one that the authors present and I consider that it would be very enriching for the work to highlight the advantages that nanobubbles present (if any) with respect to other types of shapes, justifying at all times by that.

- Several grammatical errors/typos can be found throughout the text. Therefore, the manuscript should be carefully revised from this point of view.

- The conclusions should highlight (also in the abstract) not only the improved but also the use of these kind of  nanomaterials.

Several grammatical errors/typos can be found throughout the text. Therefore, the manuscript should be carefully revised from this point of view.

Round 2

Reviewer 1 Report

Accept in current form.

Reviewer 3 Report

All the suggested corrections incorporated.